# Twenty Significant Problems in AI Research, with Potential Solutions via the SP Theory of Intelligence and Its Realisation in the SP Computer Model

**J. Gerard Wolff** 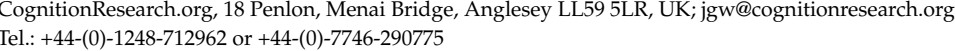

CognitionResearch.org, 18 Penlon, Menai Bridge, Anglesey LL59 5LR, UK; jgw@cognitionresearch.org; Tel.: +44-(0)-1248-712962 or +44-(0)-7746-290775

**Abstract:** This paper highlights 20 significant problems in AI research, with potential solutions via the *SP Theory of Intelligence* (SPTI) and its realisation in the *SP Computer Model*. With other evidence referenced in the paper, *this is strong evidence in support of the SPTI as a promising foundation for the development of human-level broad AI, aka artificial general intelligence*. The 20 problems include: the tendency of deep neural networks to make major errors in recognition; the need for a coherent account of generalisation, over- and under-generalisation, and minimising the corrupting effect of 'dirty data'; how to achieve one-trial learning; how to achieve transfer learning; the need for transparency in the representation and processing of knowledge; and how to eliminate the problem of catastrophic forgetting. In addition to its promise as a foundation for the development of AGI, the SPTI has potential as a foundation for the study of human learning, perception, and cognition. And it has potential as a foundation for mathematics, logic, and computing.

**Keywords:** artificial intelligence; SP Theory of Intelligence; information compression; deep neural networks

## 1. Introduction

This paper is about 20 significant problems in Artificial Intelligence (AI) research, with potential solutions via the *SP Theory of Intelligence* (SPTI) and its realisation in the *SP Computer Model* (SPCM).

The first 17 of those 20 problems in AI research have been described by influential experts in AI in interviews with science writer Martin Ford, and reported in Ford's book *Architects of Intelligence* [1].

In conjunction with other evidence for strengths of the SPTI in AI and beyond (Appendix B), the potential of the SPTI to solve those 20 problems in AI research is strong evidence in support of the SPTI as a promising foundation for the development of human-level broad AI, aka 'artificial general intelligence' (AGI).

### 1.1. The Potential of the SPTI as Part of the Foundation for Each of Four Different Disciplines

This research has potential as a foundation for four different disciplines:

- *Artificial general intelligence*. As noted above, this paper describes strong evidence in support of the SPTI as a promising foundation for the development of AGI (FDAGI). There is much more evidence in [2] and in a shortened version of that book, [3].
- *Mainstream computing*. There is evidence that the SPTI, at least when is more mature, is or will be Turing complete, meaning that it can be used to simulate any Turing machine ([2], Chapter 2). And, of course, the SPTI has strengths in AI which are largely missing from the Turing machine concept of computing, as evidenced by Alan Turing's own research on AI [4,5].
  Since the SPTI works entirely via the compression of information, the evidence, just mentioned, that it is or will be Turing complete, implies—contrary to how computing

is normally understood—that all kinds of computing may be achieved via IC. There is potential in this idea for a radically-new approach to programming and software engineering, including the potential for the automation or semi-automation of aspects of software development, and a dramatic simplification in the current plethora of programming languages and systems.

- *Mathematics, logic, and computing*. Since mathematics has been developed as an aid to human thinking, and since mathematics is the product of human minds, and in view of evidence for the importance of IC in HLPC (Appendix B.4), it should not be surprising to find that much of mathematics, perhaps all of it, may be understood as a set of techniques for IC and their application (Appendix B.5).

  In keeping with what has just been said, similar things may be said about logic and (as above) computing Section 7 in [6].

- *Human learning, perception, and cognition*. In view of evidence for the importance of IC in HLPC (Appendix B.4), the central importance of IC in the SPTI (Appendix A.1), the way in which the SPTI models several features of human intelligence (Appendix B.1), and the biological foundations for this research (Appendix B.7), suggest that the SPTI has potential as a foundation for the study of HLPC.

In addition to these four areas of study, the SPTI has potential benefits in many and perhaps all areas of science, as described in the draft paper [7].

### *1.2. Presentation*

To give readers an understanding of the SPTI, Appendix A presents a high-level view of the system, with pointers to fuller sources of information about the SPTI. Readers not already familiar with the SPTI should read this appendix, and perhaps some of those other sources, before reading the rest of the paper.

Each of the 20 sections that follow the Introduction—Sections 2 to 21 inclusive—describes one of the significant problems in AI research mentioned above, and how each one may be solved via the SPTI. In many cases, there is a demonstration to help clarify what has been said.

The appendices are not part of the substance of the paper. Apart from the abbreviations and the definitions of terms (below), they describe already-published background information that are intended to help readers understand the main substance of the paper in Sections 2–21.

The abbreviations are defined in Abbreviations and also where they are first used in the main sections of the paper or in the appendices.

Terms used in the paper are listed in Appendix C. For each one, there is a link to a section where the term is defined. In some cases, there is another link to an appendix where the term is defined.

## 2. The Need to Bridge the Divide between Symbolic and Sub-Symbolic Kinds of AI

This section is the first of those mentioned above, each of which describes a significant problem in AI research, and the potential of the SPTI to solve it.

> "Many people will tell a story that in the early days of AI we thought intelligence was symbolic, but then we learned that was a terrible idea. It didn't work, because it was too brittle, couldn't handle noise and couldn't learn from experience. So we had to get statistical, and then we had to get neural. I think that's very much a false narrative. The early ideas that emphasize the power of symbolic reasoning and abstract languages expressed in formal systems were incredibly important and deeply right ideas. I think it's only now that we're in the position, as a field, and as a community, to try to understand how to bring together the best insights and the power of these different paradigms." Josh Tenenbaum ([1], pp. 476–477).

The SPTI provides a framework that is showing promise in bridging the divide between symbolic and sub-symbolic AI:

- The concept of *SP-symbol* in the SPTI (Appendix A) can represent a relatively large 'symbolic' kind of thing such as a word, or it can represent a relatively fine-grained kind of thing such as a pixel in an image.
- The concept of SP-multiple-alignment (SPMA, see Appendix A.3) can facilitate the seamless integration of diverse kinds of knowledge (see Appendix B.1.4), and that facilitation extends to the seamless integration of symbolic and sub-symbolic kinds of knowledge.
- The SPTI has IC as a unifying principle (Appendix A.1), a principle which embraces both symbolic and sub-symbolic kinds of knowledge.

## 3. The Tendency of Deep Neural Networks to Make Large and Unexpected Errors in Recognition

"In [a recent] paper [8], [the authors] show how you can fool a deep learning system by adding a sticker to an image. They take a photo of a banana that is recognized with great confidence by a deep learning system and then add a sticker that looks like a psychedelic toaster next to the banana in the photo. Any human looking at it would say it was a banana with a funny looking sticker next to it, but the deep learning system immediately says, with great confidence, that it's now a picture of a toaster." Gary Marcus ([1], p. 318).

Although 'deep neural networks' (DNNs) often do well in the recognition of images and speech, they can make surprisingly big and unexpected errors in recognition, as described in the quote, above.

For example, a DNN may correctly recognise a picture of a car but may fail to recognise another slightly different picture of a car which, to a person, looks almost identical [9].

It has been reported that a DNN may assign an image with near certainty to a class of objects such as 'guitar' or 'penguin', when people judge the given image to be something like white noise on a TV screen or an abstract pattern containing nothing that resembles a guitar or a penguin or any other object [10].

From experience with the SPCM to date, and because of the transparency of the SPCM in both the representation and processing of knowledge (Section 10), it seems very unlikely that the SPTI, now or when it is more mature, will be vulnerable to the kinds of mistakes made by DNNs.

*Demonstrations of the SPCM'S Robustness*

The transparency of the SPCM just mentioned in both the representation and processing of knowledge (Section 10) means that all aspects of the workings of the SPCM are clear to see. Because of that transparency, some of which is illustrated in Figure 7, one can see that there is unlikely to be anything in the SPCM that would cause the kinds of haphazard errors seen in DNNs.

There is also indirect evidence for the robustness of the SPCM via its strengths in:

- *Generalisation via unsupervised learning (Section 6.1)*. There is evidence (described in Section 6.1) that the SPCM, and earlier models that learn via IC, can, via unsupervised learning, develop intuitively 'correct' SP-grammars for corresponding bodies of knowledge despite the existence of 'dirty data' in the input data. Hence, the SPCM reduces the corrupting effect of any errors in the input data.
- *Generalisation via perception (Section 6.2)*. The SPCM demonstrates an ability to parse a sentence in a manner that is intuitively 'correct', despite errors of omission, addition, and substitution in the sentence that is to be parsed (see Section 6.2). Again, the SPCM has a tendency to correct errors rather than introduce them.

## 4. The Need to Strengthen the Representation and Processing of Natural Languages

"... I think that many of the conceptual building blocks needed for human-like intelligence [AGI] are already here. But there are some missing pieces. One of them is a clear approach to how natural language can be understood to produce

knowledge structures upon which reasoning processes can operate." Stuart J. Russell ([1], p. 51).

DNNs can do well in recognising speech [11]. Also, they can produce impressive results in the translation of natural languages (NLs) using a database of equivalences between surface structures that have been built up via human mark up and pattern matching [12].

Despite successes like these, DNNs are otherwise weak in NL processing:

"... with all their impressive advantages, neural networks are not a silver bullet for natural-language understanding and generation. ... [In natural language processing] the core challenges remain: language is discreet and ambiguous, we do not have a good understanding of how it works, and it is not likely that a neural network will learn all the subtleties on its own without careful human guidance. ... The actual performance on many natural language tasks, even low-level and seemingly simple ones ... is still very far from being perfect." Yoav Goldberg ([13], Section 21.2).

By contrast, the SPCM does well in the representation and processing of NL, as outlined in Section 4.1, next.

### 4.1. Demonstrations of the SPCM's Strengths in the Processing of Natural Language

The following subsections reference demonstrations of strengths of the SPCM in the processing of NL.

The examples are all from English. But the features that are demonstrated are, almost certainly, found in all NLs. Hence, they are evidence for generality in how the SPCM may process NLs.

### 4.1.1. Parsing via SP-Multiple-Alignment

An example of the parsing of NL via the SPMA construct is shown and discussed in Appendix A.3.

### 4.1.2. Discontinuous Dependencies

SPMAs may also represent and process discontinuous syntactic dependencies in NL such as the dependency between the 'number' (singular or plural) of the subject of a sentence, and the 'number' (singular or plural) of the main verb, and that dependency may bridge arbitrarily large amounts of intervening structure ([3], Section 8.1).

In Figure A3, a number dependency (plural) is marked in row 8 by the SP-pattern 'Num PL ; NPp VPp'. Here, the SP-Symbol 'NPp' marks the noun phrase as plural, and the SP-symbol 'VPp' marks the verb phrase as plural. As is required for the marking of discontinuous dependencies, this method can mark dependencies that bridge arbitrarily large amounts of intervening structure.

In a similar way, the SPMA concept may mark the gender dependencies (masculine or feminine) within a sentence in French, and, within one sentence, number and gender kinds of dependency may overlap without interfering with each other, as can be seen in ([2], Figure 5.8 in Section 5.4.1).

### 4.1.3. Discontinuous Dependencies in English Auxiliary verbs

The same method for encoding discontinuous dependencies in NL serves very well in encoding the intricate structure of such dependencies in English auxiliary verbs. How this may be done is described and demonstrated with the SPCM in ([3], Section 8.2) and ([2], Section 5.5).

### 4.1.4. Parsing Which Is Robust against Errors of Omission, Addition, and Substitution

As can be seen in Figure 4, and described in Section 6.2, the SPCM has robust abilities to arrive at an intuitively 'correct' parsing despite errors of omission, addition, and sub-

stitution, in the sentence being parsed. Naturally, there is a limit to how many errors can be tolerated.

### 4.1.5. The Representation and Processing of Semantic Structures

In case Figures A3 and 4 have given the impression that the SPCM is only good for the processing of NL syntax, it is clear that: apart from the representation of syntax, the SPMA concept has strengths and potential in the representation and processing of other kinds of AI-related knowledge that may serve as semantics, summarised in Appendix B.1.3; and any one of those kinds of knowledge may serve as semantics in the processing of NL.

### 4.1.6. The Integration of Syntax and Semantics

Because of the versatility of SP-patterns with the SPMA concept (Appendix B.1), there is one framework which lends itself to the representation and processing of both the syntax and semantics of NLs. In addition, there is potential for the seamless integration of syntax with semantics (Appendix B.1.4). That integration means that surface forms may be translated into meanings, and *vice versa*. Examples showing how this can be done may be seen in ([2], Section 5.7, Figures 5.18 and 5.19).

### 4.1.7. One Mechanism for Both the Parsing and Production of NL

A neat feature of the SPTI is that the production of NL may be achieved by the application of IC, using *exactly* the same mechanisms as are used for the parsing or understanding of NL. How this is possible is explained in ([3], Section 4.5).

## 5. Overcoming the Challenges of Unsupervised Learning

"Until we figure out how to do this unsupervised/self-supervised/predictive learning, we're not going to make significant progress because I think that's the key to learning enough background knowledge about the world so that common sense will emerge." Yann Lecun ([1], p. 130).

From this quote it can be seen that the development of unsupervised learning is regarded as an important challenge for AI research today. So it should be of interest that: unsupervised learning in the SPCM is based on an earlier programme of research on the unsupervised learning of language [14]; unsupervised learning is a key part of the SPCM now; and it is a key part of developments envisaged for the future ([15], Section 12).

In the SP programme of research, unsupervised grammatical inference is regarded as a paradigm or framework for other kinds of unsupervised learning, not merely the learning of syntax. In addition to the learning of syntactic structures ([15], Section 12.3), it may, for example, provide a model for the unsupervised learning of non-syntactic semantic structures ([15], Section 12.1), and for learning the integration of syntax with semantics ([15], Section 12.3).

With further development, unsupervised learning in the SPCM may itself be a good foundation for other kinds of learning, such as learning by being told, learning by imitation, learning via rewards and punishments, and so on.

### 5.1. Outline of Unsupervised Learning in the SPCM

As noted in Appendix A.4, unsupervised learning in the SPCM means processing a set of New SP-patterns to discover one or more 'good' *SP-grammars*, where a 'good' SP-grammar is a set of Old SP-patterns that provide a means of encoding the given set of New SP-patterns in an economical manner.

How unsupervised learning is done in the SPCM is described quite fully in ([3], Section 5) and more fully in ([2], Chapter 9). In outline, unsupervised learning is achieved as shown in Figure 1.

```
unsupervised-learning()
{
    1 Read a set of SP-patterns into New. Old is initially empty.
    2 Compile an alphabet of alphabetic SP-symbol types in New and,
        for each type, find its frequency of occurrence and the
        number of bits required to encode it.
    3 While (there are unprocessed SP-patterns in New)
    {
        3.1 Identify the first or next SP-pattern from New as the
            'current SP-pattern from New'.
        3.2 Apply the function CREATE-SP-MULTIPLE-ALIGNMENTS() to
            create SPMAs, each one between the current SP-pattern
            from New and one or more SP-patterns from Old.
        3.3 During 3.2, the current SP-pattern from New is copied into Old,
            one symbol at a time, in such a way that the current SP-pattern
            from New can be aligned with its copy but that any one
            SP-symbol in the current SP-pattern from New cannot be aligned
            with the corresponding SP-symbol in the copy.
        3.4 Sort the SPMAs formed by this function in order
            of their compression scores and select the best
            few for further processing.
        3.5 Process the selected multiple alignments with the function
            DERIVE-SP-PATTERNS(). This function derives encoded
            SP-patterns from SPMAs and adds them to Old.
    }

    4 Apply the function SIFTING-AND-SORTING() to create one or
        more alternative SP-grammars for the SP-patterns in New, each
        one scored in terms of minimum length encoding principles.
        Each SP-grammar is a subset of the SP-patterns in Old.
}
```

**Figure 1.** The organisation of unsupervised learning in the SPCM. The workings of the functions CREATE-SP-MULTIPLE-ALIGNMENTS(), DERIVE-SP-PATTERNS() and SIFTING-AND-SORTING() are explained in ([2], Chapter 9). Adapted from Figure 9.1 in [2].

### 5.2. Demonstrations of Unsupervised Learning with the SPCM

There are examples of unsupervised learning via the SPCM in ([2], Chapter 9).

As it stands now, the SPCM can abstract words from an unsegmented body of English-like artificial language, it can learn syntactic classes of words, and it can learn the abstract structure of sentences. Its main weakness at present is in learning intermediate levels of structure such as phrases and clauses. However, it appears that such shortcomings can be overcome.

### 6. The Need for a Coherent Account of Generalisation

> "The theory [worked on by Roger Shepard and Joshua Tenenbaum] was of how humans, and many other organisms, solve the basic problem of generalization, which turned out to be an incredibly deep problem. ... The basic problem is, how do we go beyond specific experiences to general truths? Or from the past to the future?" Joshua Tenenbaum ([1], p. 468).

An important issue in unsupervised learning is how to generalise 'correctly' from the specific information which provides the basis for learning, without over-generalisation (aka 'under-fitting') or under-generalisation (aka 'over-fitting'). An associated issue is how to minimise the corrupting effect of 'dirty data', meaning data that contains errors with respect to the language or other knowledge which is being learned.

This generalisation issue is discussed in ([3], Section 5.3). The main elements of the SPTI solution are described here.

In the SP Theory of Intelligence, generalisation may be seen to occur in two aspects of AI: as part of the process of unsupervised learning; and as part of the process of parsing or recognition. Those two aspects are considered in the following two subsections.

*6.1. Generalisation via Unsupervised Learning*

The generalisation issue arises quite clearly in considering how a child learns his or her native language(s), as illustrated in Figure 2.

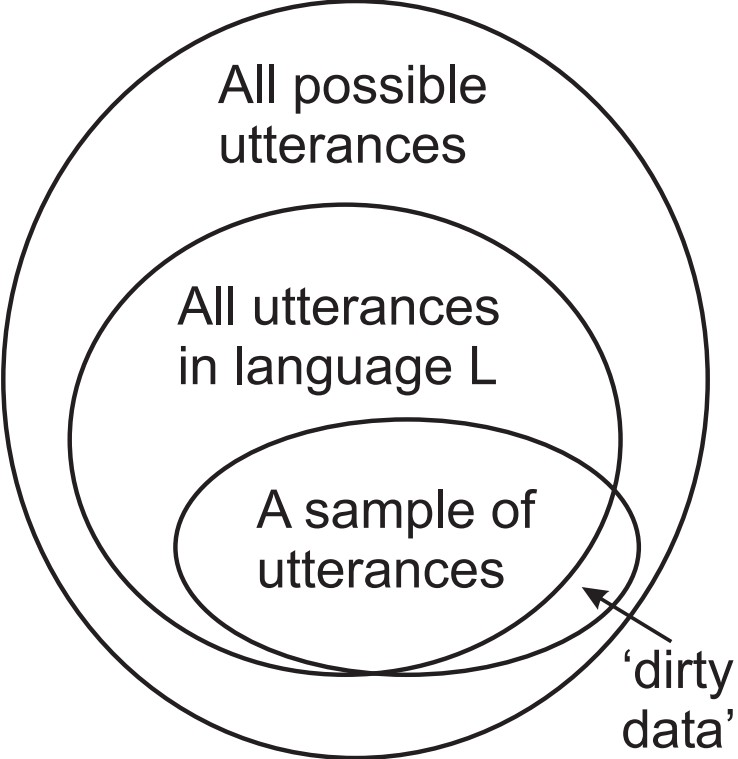

**Figure 2.** Categories of utterances involved in the learning of a first language, **L**. In ascending order of size, they are: the finite sample of utterances from which a child learns; the (infinite) set of utterances in **L**; and the larger (infinite) set of all possible utterances. Adapted from Figure 7.1 in [14], with permission.

Each child learns a language **L** from a sample of things that they hear being said by people around them. Although the sample is normally large, it is nevertheless, finite. (It is assumed here that the learning we are considering is unsupervised. This is because of evidence that, although correction of errors by adults may be helpful, children are capable of learning a first language without that kind of error correction [16,17].) That finite sample is shown as the smallest envelope in Figure 2. 'Dirty data', meaning data containing errors, is discussed below.

The variety of possible utterances in the language **L** is represented by the next largest envelope in the figure. The largest envelope represents the variety of all possible utterances, both those in **L** and everything else including grunts, gurgles, false starts, and so on.

What follows is a summary of what is already largely incorporated in the SPCM:

1.  Unsupervised learning in the SP Theory of Intelligence may be seen as a process of compressing a body of information, **I**, to achieve lossless compression of **I** into a structure **T**, where the size of **T** is at or near the minimum that may be achieved with the available computational resources.
2.  **T** may be divided into two parts:

    *   An *SP-grammar* of **I** called (**G**). Provided the compression of **I** has been done quite thoroughly, **G** may be seen to be a theory of **I** which generalises 'correctly' beyond **I**, without either over- or under-generalisations.
    *   An *encoding* of **I** in terms of **G**, called **E**. In addition to being an encoding of **I**, **E** contains all the information in **I** which occurs only once in **I**, and that is likely to include all or most of the 'dirty data' in **I**, illustrated in Figure 2.

3.   Discard **E** and retain **G**, for the following reasons:

- **G** is a distillation of what is normally regarded as the most interesting parts of **I** which should be retained.
- The encodings in **E** are not normally of much interest, and **E** is likely to contain all the 'dirty data' in **I** which may be discarded.

Demonstrations Relating to Generalisation via Unsupervised Learning

Informal tests with the SPCM ([2], Chapter 9), suggest that the SPCM can learn what are intuitively 'correct' structures, in spite of being supplied with data that is incomplete in the sense that generalisations are needed to produce a 'correct' result.

For example, the SPCM has been run with an input of New SP-patterns comprising eight sentences like 't h a t b o y r u n s' and 's o m e g i r l w a l k s', and so on, created via this framework: [(t h a t) or (s o m e)][(b o y) or (g i r l)][(r u n s) or w a l k s)].

With that input, the best SP-grammar created by the SPCM is shown in Figure 3. (As can be seen in the example in Figure 3, it was created by a version of the SPCM that used the characters '<' and '>' to mark the beginning and end of each SP-pattern. This led to unreasonable complications in the SPCM and has now been dropped in favour of SP-symbols like 'N' and '#N' which could be processed like all other SP-symbols.) In terms of our intuitions, this is at or near the 'correct' result for the given input.

```
< %2 2 s o m e >
< %2 3 t h a t >
< %1 5 b o y >
< %1 6 g i r l >
< %3 4 r u n s >
< %3 7 w a l k s >
< 1 < %2 > < %1 > < %3 > >
```

**Figure 3.** The best SP-grammar found by the SPCM when New contains the eight sentences described in the text.

Now for generalisation via unsupervised learning: if the program is run again with the same input, but *without* the sentence '(t h a t g i r l r u n s)', the SP-grammar that is created is *exactly* the same as shown in Figure 3. In other words, the SPCM has generalised from the data and produced a result which, in terms of our intuitions, is correct.

Of course, simple examples like the one just described are only a beginning, and it will be interesting to see how the SPCM generalises with more ambitious examples, especially when the program has reached the stage when it can produce plausible SP-grammars from samples of NL.

Even now, there is a reason to have confidence in the model of generalisation that has been described: because the model's basis in compression of information is consistent with much other evidence for the significance of IC in the workings of brains and nervous systems [18].

*Dirty data*. With regard to dirty data, mentioned above and shown in Figure 2: in informal experiments with models of language learning developed in earlier research that have IC as a unifying principle: "In practice, the programs MK10 and SNPR have been found [to produce intuitively 'correct' results but] to be quite insensitive to errors (of omission, commission, or substitution) in their data." ([14], p. 209).

### 6.2. Generalisation via Perception

The SPCM has a robust ability to recognise things or to parse NL despite errors of omission, addition, or substitution in what is being recognised or parsed. Incidentally, the assumption here is that recognition in any sensory modality may be understood largely as parsing, as described in ([19], Section 4).

Demonstration of Generalisation via Perception

The example here makes reference, first, to Figure A3 in Appendix A.3.2, which shows how an SPMA can achieve the effect of parsing the sentence 't w o k i t t e n s p l a y' in terms of grammatical categories, including words.

To illustrate generalisation via perception, Figure 4, below, shows how the SPCM, with a New SP-pattern that contains errors ('t o k i t t e m s p l a x y'), may achieve what is intuitively a 'correct' analysis of the sentence despite the errors, which are described in the caption of the figure.

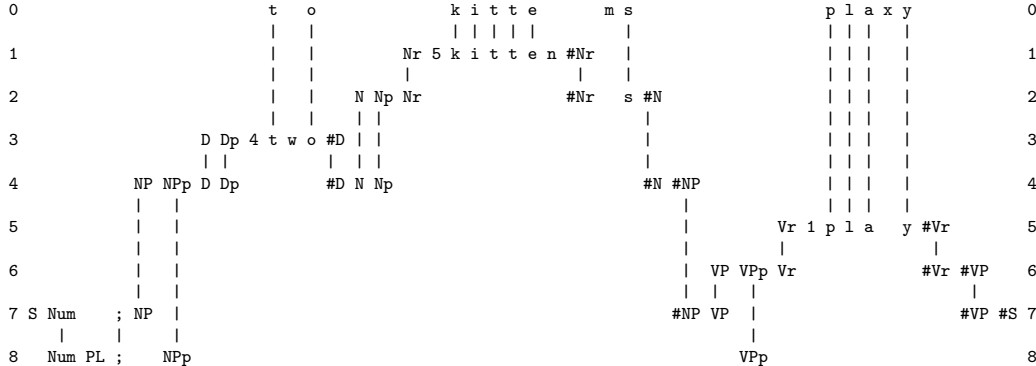

**Figure 4.** As in Figure A3 but with errors in the sentence in row 0 ('t o k i t t e m s p l a x y'): an error of omission ('t o' instead of 't w o'), an error of substitution ('k i t t e m s' instead of 'k i t t e n s'), and an error of addition ('p l a x y' instead of 'p l a y'). Adapted from Figure 2 in [20], with permission. This figure was originally published in *Data & Knowledge Engineering*, 60, J. G. Wolff, "Towards an intelligent database system founded on the SP Theory of Computing and Cognition", 596–624, Copyright Elsevier (2007).

Recognition in the face of errors, as illustrated in Figure 4, may be seen as a kind of generalisation, where an incorrect form is generalised to the correct form.

## 7. How to Learn Usable Knowledge from a Single Exposure or Experience

"How do humans learn concepts not from hundreds or thousands of examples, as machine learning systems have always been built for, but from just one example? . . . Children can often learn a new word from seeing just one example of that word used in the right context, . . . You can show a young child their first giraffe, and now they know what a giraffe looks like; you can show them a new gesture or dance move, or how you use a new tool, and right away they've got it . . . " Joshua Tenenbaum ([1], p. 471).

Most DNNs incorporate some variant of the idea that, in learning, neural connections gradually gain strength, either via 'backpropagation' or via some variant of Donald Hebb's concept of learning, which may be expressed briefly as "Neurons that fire together, wire together."

This gradualist model of learning in DNNs seems to reflect the way that it takes time to learn a complex skill such as playing the piano well, or competition-winning abilities in pool, billiards, or snooker.

But the gradualist model conflicts with the undoubted fact that people can and often do learn usable knowledge from a single occurrence or experience: memories for significant events that we experience only once may be retained for many years.

For example, if a child touches something hot, he or she is likely to retain what they have learned for the rest of their lives, without the need for repetition.

One-trial learning accords with our experience in any ordinary conversation between two people. Normally, each person responds immediately to what the other person has said.

It is true that DNNs require the taking in of information supplied by the user, and may thus be said to have learned something from a single exposure or experience. But unlike the SPTI, that knowledge is not useful immediately. Any information taken in by a DNN at the beginning of a training session does not become useful until much more information has been taken in, and there has been a gradual strengthening of links within the DNN. Here there is a clear advantage of the SPTI compared with DNNs. The SPCM can use New information immediately.

It is perhaps worth mentioning that 'one-trial learning' is different from 'zero-shot learning', as for example, in the 'CLIP' system that features in [21,22]. 'Zero-shot learning' is:

> "... a problem setup in machine learning, where at test time, a learner observes samples from classes, which were not observed during training, and needs to predict the class that they belong to. Zero-shot methods generally work by associating observed and non-observed classes through some form of auxiliary information, which encodes observable distinguishing properties of objects. ... For example, given a set of images of animals to be classified, along with auxiliary textual descriptions of what animals look like, an artificial intelligence model which has been trained to recognize horses, but has never been given a zebra, can still recognize a zebra when it also knows that zebras look like striped horses." 'Zero-shot learning', *Wikipedia*, tinyurl.com/yyybvm8x, accessed on 29 August 2022.

### 7.1. Demonstration of One-Trial Learning

There is a demonstration of one-trial learning in the parsing of a New SP-pattern representing a sentence (fresh from the system's environment) in the example shown in Figure A3 (Appendix A.3.1).

In the example, the New SP-pattern has been 'learned' in much the same way that a tape recorder may be said to 'learn' sounds, or a camera may be said to 'learn' images.

What is different from a tape recorder or camera is that the newly 'learned' information may be used immediately in, for example, parsing (as illustrated in Figure A3 or in any of the other aspects of intelligence summarised in Appendix B.1).

### 7.2. Slow Learning of Complex Knowledge or Skills

In addition to the explanation which the SPTI provides for one-trial learning, the SPTI also provides an explanation for why people are relatively slow at learning complex bodies of knowledge or complex skills like those mentioned earlier.

It seems likely that the slow learning of complex things is partly because there is a lot to be learned, and partly because that kind of learning requires a time-consuming search through a large abstract space of ways in which the knowledge may be structured in order to compress it and thus arrive at an efficient configuration.

In the SPCM, that kind of slow learning occurs when the mechanisms of unsupervised learning are needed to sift and sort through the many ways in which knowledge may be structured to achieve good levels of IC, as for example, in the induction of an efficient SP-grammar from raw linguistic data (Section 5.2).

Although such learning in the SPCM is relatively slow compared with one-trial learning by the model, it is likely, in mature versions of the SPCM, to prove to be relatively efficient and fast compared with the large computational resources needed by DNNs (see Section 9).

## 8. How to Achieve Transfer Learning

> "We need to figure out how to think about problems like transfer learning, because one of the things that humans do extraordinarily well is being able to learn something, over here, and then to be able to apply that learning in totally new environments or on a previously unencountered problem, over there." James Manyika ([1], p. 276).

Transfer learning is fundamental in the SPCM. This is because the system does not suffer from catastrophic forgetting (Section 21). This means that anything that has been learned and stored in the repository of Old SP-patterns is available at any time to be incorporated in any other structure.

With DNNs, it is possible, to some extent, to sidestep the problem of catastrophic forgetting by making a copy of a DNN that has already learned something (eg, how to recognise a domestic cat) and then exposing the copy to images of things that are related to cats, such as lions (see, for example, ([23], Section 2)). Then the prior knowledge of cats may facilitate the later learning of what lions look like.

A recent survey of research on transfer learning [24], mainly about research with DNNs, concludes that "Several directions are available for future research in the transfer learning area. ... And new approaches are needed to solve the knowledge transfer problems in more complex scenarios." (p. 71)

From this conclusion, and from the quotation at the beginning of this section, it is clear that transfer learning in DNNs falls far short of what is achieved by transfer learning in the SPCM—where transfer learning is an integral part of how the system works, so that new concepts can be created from any combination of New and Old information.

*Demonstration of Transfer Learning with the SPCM*

Here is a simple example of how the SPCM can achieve transfer learning between one Old SP-pattern and one New SP-pattern:

- At the beginning, there is one Old SP-pattern already stored, namely: '< %1 3 t h a t b o y r u n s >'.
- Then a New SP-pattern is received: 't h a t g i r l r u n s'.
- The best SP-multiple alignment for these two SP-patterns is shown in Figure 5.
- From that SPMA, the SPCM derives SP-patterns as shown in Figure 6. This is the beginnings of an SP-grammar for sentences of a given form.
- Because IC in the SPCM is always lossless, the SP-grammar in Figure 6 generates the original two SP-patterns from which the SP-grammar is derived.

```
0       t h a t g i r l r u n s   0
        | | | |         | | | |
1 < %1 3 t h a t b o y   r u n s > 1
```

**Figure 5.** The best SPMA created by the SPCM between the Old SP-pattern in row 1 and the New SP-pattern in row 0.

```
< %1 1 t h a t >
< %2 2 r u n s >
< %3 3 b o y >
< %3 4 g i r l >
< 5 < %1 > < %3 > < %2 > >
```

**Figure 6.** The SP-grammar created by the SPCM from the SPMA shown in Figure 5.

This simple example is just a taste of how the SPCM works. As mentioned above, transfer learning is an integral part of the SPCM, something that could not be removed from the system without completely destroying its generality and power in diverse aspects of intelligence.

## 9. How to Increase the Speed of Learning, and Reduce Demands for Large Amounts of Data and for Large Computational Resources

"[A] stepping stone [towards artificial general intelligence] is that it's very important that [AI] systems be a lot more data-efficient. So, how many examples do you need to learn from? If you have an AI program that can really learn from a single example, that feels meaningful. For example, I can show you a new object, and you look at it, you're going to hold it in your hand, and you're thinking,

'I've got it.' Now, I can show you lots of different pictures of that object, or different versions of that object in different lighting conditions, partially obscured by something, and you'd still be able to say, 'Yep, that's the same object.' But machines can't do that off of a single example yet. That would be a real stepping stone to [artificial general intelligence] for me." Oren Etzioni ([1], p. 502).

In connection with the large volumes of data and large computational resources that are often associated with the training of DNNs, it has been discovered by Emma Strubell and colleagues [25] that, when electricity is generated from fossil fuels, the process of training a large AI model can emit more than 626,000 pounds of carbon dioxide, which is equivalent to nearly five times the lifetime emissions of the average American car, including the manufacture of the car itself.

The SPTI suggests two main ways in which learning can be done faster, with less data, and fewer computational resources:

- *Learning via a single exposure or experience*. Take advantage of the way in which the SPCM can, as a normal part of how it works, learn usable knowledge from a single exposure or experience (Section 7).
- *Transfer learning*. Take advantage of the way in which the SPCM can, and frequently does, incorporate already-stored knowledge in the learning of something new (Section 8).

It is true that the SPTI is likely, like people, to be relatively slow in learning complex knowledge and skills (Section 7.2). But even here there can be benefits from one-trial learning and transfer learning.

It seems likely that in unsupervised learning, a mature SPCM will be substantially more efficient than the current generation of DNNs.

## 10. The Need for Transparency in the Representation and Processing of Knowledge

"The current machine learning concentration on deep learning and its non-transparent structures is such a hang-up." Judea Pearl ([1], p. 369).

It is now widely recognised that a major problem with DNNs is that the way in which learned knowledge is represented in such systems is far from being comprehensible by people, and likewise for the way in which DNNs arrive at their conclusions.

These deficiencies in DNNs are of concern for reasons of cost, safety, legal liability, fixing problems in systems that use DNNs, and perhaps more.

By contrast, with the SPCM:

- All knowledge in the SPCM is represented transparently by SP-patterns, in structures, some of which are likely to be familiar to people such as part-whole hierarchies, class-inclusion hierarchies, and more (see ([6], Section 5).
- There is a comprehensive audit trail for the creation of each SPMA. The structure of one such audit trail is shown in Figure 7.
- There is also a comprehensive audit trail for the learning of SP-grammars by the SPCM.

There is a fairly full discussion of issues relating to transparency in the representation and processing of knowledge in [26].

*Demonstrations Relating to Transparency*

Figure 7 shows how the SPMA in Figure A3 was created. That figure should be interpreted as described in its caption. It illustrates some but not all of the transparency of the SPCM.

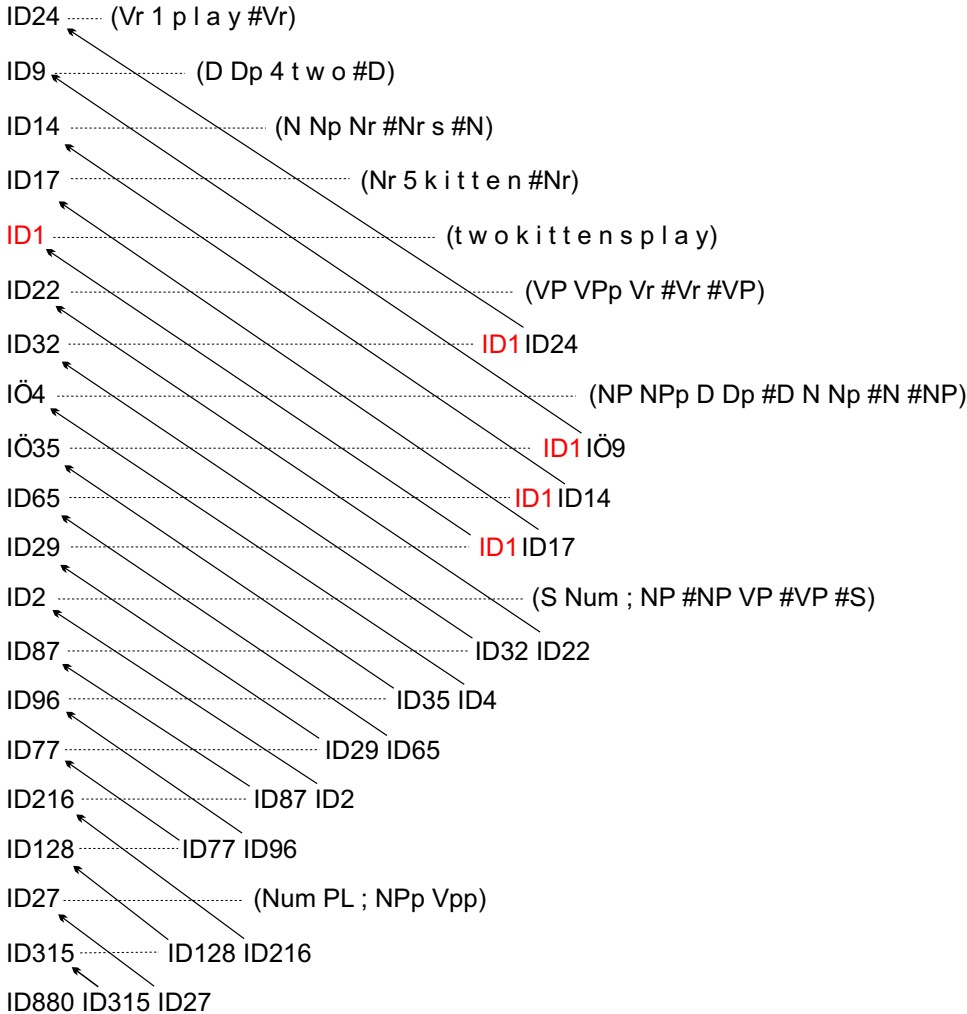

**Figure 7.** An audit trail for the creation of the SPMA shown in Figure A3. The line of text at the bottom of the figure shows, on the left, the identification number for the SPMA shown in Figure A3. To the right of that number are IDs of the two SPMAs from which the SPMA was derived. From each of latter two IDs, there is an arrow pointing to the same ID at the beginning of a line above. As before, this ID represents an SPMA, and to the right of that ID are two more IDs representing the two SPMAs from which the main ID for that line was derived. The remaining lines in the figure are the same except that, at higher levels, we begin to meet IDs that represent Old SP-patterns supplied to the SPCM at the beginning of processing. With the identifier 'ID1', there are five instances in the figure (each one shown in red), so to avoid undue clutter, only one arrow is shown. Reproduced from Figure 6 in [26], with permission.

The partial SPMAs themselves are not shown in the figure but they are shown in the full audit trail from which the figure was derived—which is too big to be shown here. That full audit trail contains much more information than what is shown in Figure 7, including measures of IC associated with each SPMA, and absolute and conditional probabilities associated with each SPMA, calculated as described in ([3], Section 4.4).

Figure 7 is only for explanation: it is not something that users of the SPCM would be required to use. In the full audit trail, users may trace the origin of any structure by following relevant pointers.

In addition to audit trails for building of SPMAs, the SPCM provides a comprehensive audit trail for all intermediate structures that are created for unsupervised learning. Even with small examples, the audit trail is much too big to be shown here.

## 11. How to Achieve Probabilistic Reasoning That Integrates with Other Aspects of Intelligence

Although this section (about probabilistic reasoning), and Section 12 (about common-sense reasoning and commonsense knowledge (CSRK), are both about reasoning, they are described separately because they are not yet well integrated.

> "What's going on now in the deep learning field is that people are building on top of these deep learning concepts and starting to try to solve *the classical AI problems of reasoning* and being able to understand, program, or plan." Yoshua Bengio ([1], p. 21), emphasis added.

A strength of the SPTI compared with DNNs is that, via the SPCM, several different kinds of probabilistic reasoning can be demonstrated, without any special provision or adaptation ([3], Section 10).

The kinds of probabilistic reasoning that can be demonstrated with the SPCM are summarised in Section B.1.2.

Owing to the central importance of IC in the SPTI (Appendix A.1), and owing to the intimate connection that is known to exist between IC and concepts of probability (see 'Algorithmic Probability Theory' developed by Solomonoff, [27,28], ([29], Chapter 4)), the SPTI is intrinsically probabilistic. With every SPMA created by the SPCM, including SPMAs produced in any of the kinds of reasoning mentioned above, absolute and relative probabilities are calculated ([3], Section 4.4).

As with the processing of NL (Section 4), a major strength of the SPTI with reasoning is that there can be seamless integration of various aspects of intelligence and various kinds of knowledge, in any combination (Appendix B.1.4).

Here, that kind of seamless integration would apply to the several kinds of probabilistic reasoning mentioned above, and other aspects of intelligence, and varied kinds of knowledge.

*Demonstrations Relating to Probabilistic Reasoning*

Examples of probabilistic reasoning by the SPCM are shown in (([3], Section 10), and there are more in ([2], Chapter 7)). These sources provide a fairly full picture of the varied kinds of probabilistic reasoning that may be achieved with the SPCM.

To give some of the flavour of how the SPCM can be applied to nonmonotonic reasoning, Figure 8 shows one of the three best SPMAs created by the SPCM with the New SP-pattern 'bird Tweety' (which appears in column 0, and which may be interpreted as 'Tweety is a bird'), and with a store of Old SP-patterns representing aspects of birds in general and also of specific kinds of birds such as ostriches and penguins. (This SPMA is arranged with SP-patterns in columns instead of rows, but otherwise the SPMA may be interpreted in exactly the same way as other SPMAs shown in this paper.)

The SPMA in Figure 8 shows that Tweety is likely to be a bird that, like most birds, is warm-blooded and has wings and feathers (column 1).

One of the other SPMAs formed at the same time (not shown) suggests that Tweety might be an ostrich, and another shows that Tweety might be a penguin.

Taking the three SPMAs together, we may conclude that: with a relative probability of 0.66, Tweety can fly; but it is possible that Tweety as an ostrich would not be able to fly ($p = 0.22$); and it is also possible that Tweety as a penguin would not be able to fly ($p = 0.12$). How these probabilities are calculated is described in ([3], Section 4.4).

If the SPCM is run again, with the New SP-pattern 'penguin Tweety' (meaning that Tweety is a penguin), and with the same store of Old SP-patterns as before, the best SPMA formed by the SPCM is shown in Figure 9. From this we can infer that Tweety would certainly *not* be able to fly ($p = 1.0$). Likewise if the New SP-pattern is 'ostrich Tweety'.

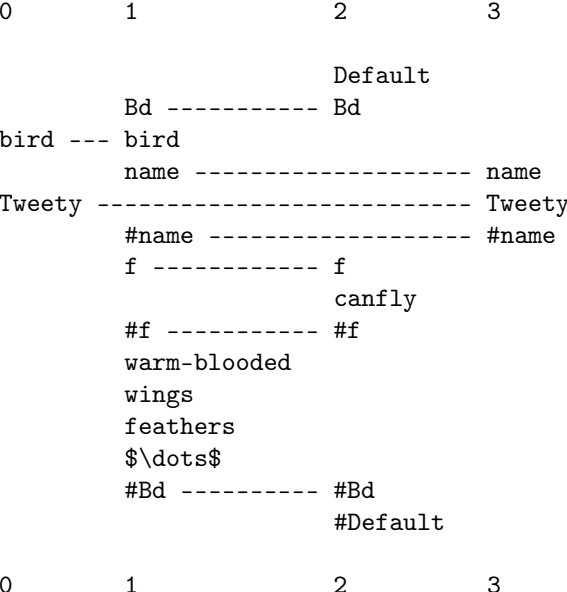

```
0           1              2            3

                          Default
           Bd ----------- Bd
bird --- bird
           name ------------------ name
Tweety ------------------------- Tweety
           #name ------------------ #name
           f ----------- f
                          canfly
           #f ----------- #f
           warm-blooded
           wings
           feathers
           $\dots$
           #Bd ---------- #Bd
                          #Default

0           1              2            3
```

**Figure 8.** One of the three best SPMAs formed by the SPCM from the New SP-pattern shown in column 0, 'bird Tweety', and a repository of Old SP-patterns describing information about birds in general and more specific kinds of birds such as ostriches and penguins. Reproduced from ([2], Figure 7.10).

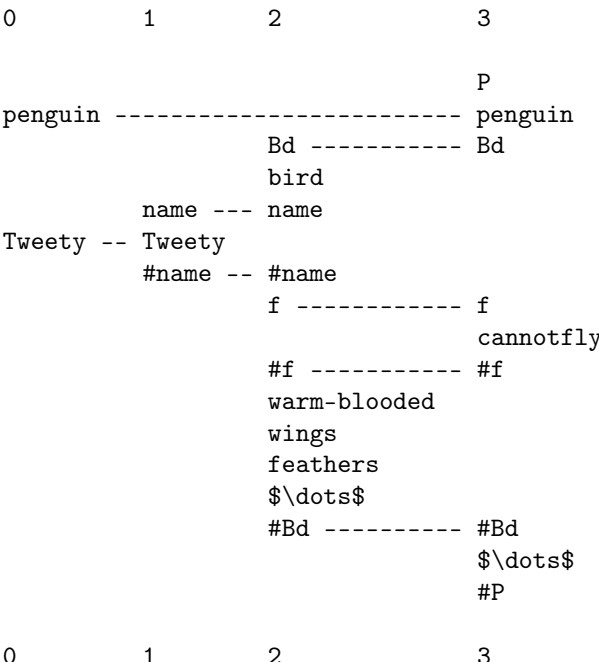

```
0           1          2            3

                                  P
penguin ------------------------- penguin
                      Bd ----------- Bd
                      bird
           name --- name
Tweety -- Tweety
           #name -- #name
                      f ----------- f
                                  cannotfly
                      #f ----------- #f
                      warm-blooded
                      wings
                      feathers
                      $\dots$
                      #Bd ---------- #Bd
                                  $\dots$
                                  #P

0           1          2            3
```

**Figure 9.** The best SPMA formed by the SPCM with the New SP-pattern 'penguin Tweety' and with SP-patterns in Old as described in the text. The relative probability of this SPMA is 1.0. Reproduced from ([2], Figure 7.12).

Thus, in accordance with the concept of nonmonotonic reasoning, the SPCM makes one set of inferences with the information that Tweety is a bird, but these inferences can be changed when we have information that, for example, Tweety is a penguin or Tweety is an ostrich.

## 12. The Challenges of Commonsense Reasoning and Commonsense Knowledge

This section is about commonsense reasoning (CSR) and commonsense knowledge (CSK), where the two together may be referred to as 'CSRK'. As noted in Section 11, although that section and this one are both about reasoning, they are described separately because they are not yet well integrated.

> "We don't know how to build machines that have human-like common sense. We can build machines that can have knowledge and information within domains, but we don't know how to do the kind of common sense we all take for granted." Cynthia Breazeal ([1], p. 456).

There appears to be little research on how DNNs might be applied in CSRK research [30] (but see [31,32]). Preliminary studies suggest that the SPTI has potential in this area [33,34].

*Demonstrations Relating to Commonsense Reasoning and Commonsense Knowledge*

Aspects of CSRK can be modelled with the SPCM ([34], Sections 4 to 6): how to interpret a noun phrase like "water bird"; how, under various scenarios, to assess the strength of evidence that a given person committed a murder; and how to interpret the horse's head scene in *The Godfather* film.

A fourth problem—how to model the process of cracking an egg into a bowl—is beyond what can be done with the SPTI as it is now ([34], Section 9). It seems likely that progress on that problem will depend on progress in the modelling of structures in two dimensions ([15], Section 9.1), and three dimensions ([15], Section 9.2), and with the time dimension as well ([15], Section 9.3).

## 13. How to Minimise the Risk of Accidents with Self-Driving Vehicles

> "... the principal reason [for pessimism about the early introduction of driverless cars for all situations is] that if you're talking about driving in a very heavy metropolitan location like Manhattan or Mumbai, then the AI will face a lot of unpredictability. It's one thing to have a driverless car in Phoenix, where the weather is good and the population is a lot less densely packed. The problem in Manhattan is that anything goes at any moment, nobody is particularly well-behaved and everybody is aggressive, the chance of having unpredictable things occur is much higher." Gary Marcus ([1], p. 321).

A naive approach to the development of self-driving vehicles would be to teach the vehicle's computer in the same way that one might teach a person who is learning to drive. This will not work unless or until the computer is equipped with human-level abilities for generalisation, and abilities to minimise the corrupting effect of dirty data, along the lines outlined in Section 6. Even then, it is probably best to build up the necessary knowledge via unsupervised learning with inputs of information about driving conditions and responses by a skilled human driver.

How those principles may be applied to the development of driverless cars is described in [35].

## 14. The Need for Strong Compositionality in the Structure of Knowledge

> "By the end of the '90s and through the early 2000s, neural networks were not trendy, and very few groups were involved with them. I had a strong intuition that by throwing out neural networks, we were throwing out something really important. ...
>
> Part of that [intuition] was because of something that we now call compositionality: The ability of these systems to represent very rich information about the data in a compositional way, where you compose many building blocks that correspond to the neurons and the layers." Yoshua Bengio ([1], pp. 25–26).

The neurons and layers of a DNN may be seen as building blocks for a concept, and may thus be seen as an example of compositionality, as described in the quote above.

But any such view of the layers in a DNN is weak, with many exceptions to 'strong' compositionality. In general, DNNs fail to capture the way in which we conceptualise a complex thing like a car in terms of smaller things (engine, wheels, etc.), and these in terms of still smaller things (pistons, valves, etc.), and so on.

In this connection, the SPTI has a striking advantage compared with DNNs. Any SP-pattern may contain SP-symbols that serve as references to other SP-patterns, a mechanism which allows part-whole hierarchies and class-inclusion hierarchies to be built up through as many levels as are required ([3], Section 9.1). At the same time, the SPMA construct is much more versatile than a system that merely builds hierarchical structures (see Appendices B.1 and B.2).

*Demonstrations of Compositionality with the SPTI*

The kind of strong compositionality which is the subject of discussion can be seen in Figure A3:

- The word 't w o' is part of the 'determiner' category as shown in the SP-pattern 'D Dp 4 t w o #D' in row 3;
- The word 'k i t t e n' is the 'root' of a noun represented by the SP-pattern 'Nr 5 k i t t e n #Nr' in row 1, and this is part of the 'noun' category represents by 'N Np Nr #Nr s #N' in row 2;
- Both 'D Dp 4 t w o #D' and 'N Np Nr #Nr s #N' are part of the 'noun phrase' structure represented by the SP-pattern 'NP NPp D Dp #D N Np #N #NP' in row 4;
- There is a similar but simpler hierarchy for the 'verb phrase' category represented by the SP-pattern 'VP VPp Vr #Vr #VP' in row 6;
- The noun phrase structure, 'NP NPp D Dp #D N Np #N #NP', and the verb phrase structure, 'VP VPp Vr #Vr #VP', are the two main components of a sentence, represented by the SP-pattern 'S Num ; NP #NP VP #VP #S' in row 7.

Apart from part-whole hierarchies like the one just described, the SPTI also lends itself to the representation and processing of class-inclusion hierarchies, as can be seen in ([3], Figure 16, Section 9.1).

## 15. Establishing the Importance of Information Compression in AI Research

There is little about IC in *Architects of Intelligence*, except for some brief remarks about autoencoders:

> "Autoencoders have changed quite a bit since [the] original vision. Now, we think of them in terms of taking raw information, like an image, and transforming it into a more abstract space where the important, semantic aspect of it will be easier to read. That's the encoder part. The decoder works backwards, taking those high-level quantities—that you don't have to define by hand—and transforming them into an image. That was the early deep learning work. Then a few years later, we discovered that we didn't need these approaches to train deep networks, we could just change the nonlinearity." Yoshua Bengio ([1], pp. 26–27).

This fairly relaxed view of IC in AI research, as described by an influential AI expert, and the rather low profile of IC in a wide-ranging review of research on DNNs ([30], Section 4.4), and in most publications about DNNs, (An exception here is [36] which describes "a theoretical framework for approximate planning and learning in partially observed system" (Abstract), with compression in its analysis, but with vastly more complexity than the treatment of IC in the SPTI [2, Section 2.2].) contrasts with the central role of IC in the SP programme of research:

1. There is good evidence that IC is fundamental in HLPC [18], and in mathematics [6].
2. IC is bedrock in the design of the SPTI ([3], Section 2.1).
3. Via the SPMA concept, IC appears to be largely responsible for the strengths and potential of the SPTI in AI-related functions (Appendix B.1), and largely responsible for potential benefits and applications of the SPTI in other areas (Appendix B.2).

With regard to the first point, Marcus provides persuasive evidence in his book *Kluge* [37], that the haphazard nature of natural selection produces kluges in human thinking, meaning clumsy makeshift solutions that nevertheless work. But, at the same time, there is compelling evidence for the importance of IC in the workings of brains and nervous systems [18]. Probably, the two ideas are both true.

## 16. Establishing the Importance of a Biological Perspective in AI Research

Since people are biological entities, this section, about the importance of a biological perspective in AI research, includes evidence from human psychology and neuroscience.

> "Deep learning will do some things, but biological systems rely on hundreds of algorithms, not just one algorithm. [AI researchers] will need hundreds more algorithms before we can make that progress, and we cannot predict when they will pop." Rodney Brooks ([1], p. 427).

What Rodney Brooks describes in this quote is much like Marvin Minsky's concept of many diverse agents as the basis for AI [38]. It seems that both views are unfalsifiable because, for every attempt to prove the theory wrong, a new algorithm can be added to plug the gap. And that is likely to mean a theory with ever-decreasing merit in terms of Ockham's razor.

### 16.1. IC and the Biological Foundations of the SPTI

By contrast with the 'many algorithms' perspectives of Brooks and Minsky, the SP programme of research has, from the beginning, been tightly focussed on the importance of IC in the workings of brains and nervous systems [18] and the corresponding importance of IC as a unifying principle in the SPTI (Appendix A.1).

Much of the evidence for the importance of IC in HLPC, presented in [18], comes from neuroscience, psychology, and other aspects of biology. Hence, the SPTI is inspired in part by evidence from biology.

### 16.2. SP-Neural and Inputs from Neuroscience

*SP-Neural* is a version of the SPTI couched in terms of neurons and their interconnections and intercommunications [39]. It provides further evidence for a biological perspective in the development of the SPTI because there has been considerable input from neuroscience in the development of SP-Neural, and in its description in [39].

## 17. Establishing Whether Knowledge in Brains or AI Systems Should Best Be Represented in 'Distributed' or 'Localist' Form

> "In a hologram, information about the scene is distributed across the whole hologram, which is very different from what we're used to. It's very different from a photograph, where if you cut out a piece of a photograph you lose the information about what was in that piece of the photograph, it doesn't just make the whole photograph go fuzzier." Geoffrey Hinton ([1], p. 79).

A persistent issue in AI and in theories of HLPC is whether or not knowledge in the brain is represented in a 'distributed' or 'localist' form, and whether the same principles should be applied in AI models. This is essentially the much-debated issue of whether the concept of 'my grandmother' is represented in one place in one's brain or whether the concept is represented via a diffuse collection of neurons throughout the brain:

- In DNNs, knowledge is distributed in the sense that the knowledge is encoded in the strengths of connections between many neurons across several layers of each DNN. Since DNNs provide the most fully developed examples of AI systems with distributed knowledge, the discussion here assumes that DNNs are representative of such systems.
- The SPTI, in both its abstract form (Appendix A) and as SP-Neural [39], is unambiguously localist.

It seems now that the weight of evidence favours a localist view of how knowledge is represented in the brain:

- Mike Page ([40], pp. 461–463) discusses several studies that provide direct or indirect evidence in support of localist encoding of knowledge in the brain.
- It is true that if knowledge of one's grandmother is contained within a neural SP-pattern, death of that neural SP-pattern would destroy knowledge of one's grand-mother. But:
    - As Barlow points out ([41], pp. 389–390), a small amount of replication will give considerable protection against this kind of catastrophe.
    - Any person who has suffered a stroke, or is suffering from dementia, may indeed lose the ability to recognise close relatives or friends.
- In connection with the 'localist' view of brain organisation, an important question is whether or not there are enough neurons in the human brain to store the knowledge that a typical person, or, more to the point, an exceptionally knowledgeable person, will have?

    Arguments and calculations relating to this issue suggest that it is indeed possible for us to store what we know in localist form, and with substantial room to spare for multiple copies ([2], Section 11.4.9). A summary of the arguments and calculations is in ([39], Section 4.4).

    Incidentally, multiple copies of a localist representation in which each copy is localist, or a neural SP-pattern for the representation of each concept, is not the same as the diffuse representation of knowledge in a distributed representation.

Assuming that, in the quest for AGI, we may maximise our chances of success by imitating the structure and workings of the brain, the case for the SPTI as a FDAGI is strengthened by evidence that the SPTI, like the brain, stores knowledge in localist form.

## 18. The Learning of Structures from Raw Data

"Evolution does a lot of architecture search; it designs machines. It builds very differently, structured machines across different species or over multiple genera-tions. We can see this most obviously in bodies, but there's no reason to think it's any different in [the way that] brains [learn]." Josh Tenenbaum ([1], p. 481).

With unsupervised learning in the SP programme of research, the aim is not merely to learn associations between things but to learn the *structures* in the world. (There are similar objectives in research on DNNs, as for example in [42].) This has been a theme of research on the learning of a first language [14] and in the SP research (([3], Section 5), ([2], Chapter 9)).

Those publications are about the unsupervised learning of structure in one-dimensional data. But other developments are envisaged within the SP programme of research. It is anticipated that the SPCM will be generalised to represent and process SP-patterns in two dimensions ([15], Section 9.1). That should provide the foundation for the unsupervised learning of parts and sub-parts of pictures and diagrams, for classes and subclasses of such entities, and for three-dimensional structures.

There is also potential for the learning and representation of: *class hierarchies* and *inheritance* ([43], Section 6.6); and for the processing of parallel streams of information ([44], Sections V-G, V-H, and V-I, and Appendix C).

*Demonstrations of the Unsupervised Learning of Structures from Raw Data*

The SPCM as it is now has demonstrated the unsupervised learning of word structure and the unsupervised learning of simple English-like SP-grammars, including classes of words and high-level sentence structure ([3], Section 5.1.1). As with the earlier work on the learning of a first language, the learning by the SPCM is achieved without any explicit clues to structure in the data from which it learns.

What has been achieved already demonstrates the clear potential of the framework for learning segmental and disjunctive (class) structures in raw data via IC which is itself achieved via the Matching and Unification of Patterns (ICMUP). Here, *unification* is simply the merging of two or more SP-patterns, or parts of such patterns, to make a single SP-pattern, or part thereof.

This is consistent with what was earlier achieved with the MK10 and SNPR computer models of the learning of segmental structure of language and the learning of grammars [14], but within the entirely new framework of the SP research.

Those developments will themselves facilitate the learning of structures in three dimensions as outlined in ([19], Sections 6.1 and 6.2). In brief, the structure of a 3D object may be created from overlapping pictures of the object from several different angles. The overlap between neighbouring pictures would provide for the stitching together of the pictures, in much the same way as is done in the creation of a panoramic view of a scene by the stitching together of overlapping pictures of the scene.

The creation of 3D representations of objects from overlapping pictures is demonstrated by businesses that do it as a service for customers. It is also demonstrated by the way in which 3D representations of streets in Google's Streetview are created from overlapping pictures.

## 19. The Need to Re-Balance Research towards Top-Down Strategies

This section and the ones that follow describe problems in AI that are not apparently considered in [1] but are significant problems in AI research that the SPTI has potential to solve.

> "The central problem, in a word: current AI is *narrow*; it works for particular tasks that it is programmed for, provided that what it encounters isn't too different from what it has experienced before. That's fine for a board game like Go—the rules haven't changed in 2,500 years—but less promising in most real-world situations. Taking AI to the next level will require us to invent machines with substantially more flexibility. ... To be sure, ... narrow AI is certainly getting better by leaps and bounds, and undoubtedly there will be more breakthroughs in the years to come. But it's also telling: AI could and should be about so much more than getting your digital assistant to book a restaurant reservation." Gary Marcus and Ernest Davis ([45], pp. 12–14), emphasis in the original.

This quote is, in effect, a call for a top-down, breadth-first strategy in AI research, developing a theory or theories that can be applied to a range of phenomena, not just one or two things in a narrow area (Appendix B.6).

In this connection, the SPTI scores well. *It has been developed with a unique top-down strategy: attempting simplification and integration across an unusually broad canvass: across AI, mainstream computing, mathematics, and HLPC.*

Here are some key features of a top-down strategy in research, and their potential benefits:

1. *Broad scope*. Achieving generality requires that the data from which a theory is derived should have a broad scope, like the overarching goal of the SP programme of research, summarised above.

2. *Ockham's razor, Simplicity and Power*. That broad scope is important for two reasons:

   - In accordance with Ockham's razor, a theory should be as *Simple* as possible but, at the same time, it should retain as much as possible of the descriptive and explanatory *Power* of the data from which the theory is derived.
   - But measures of Simplicity and Power are more important when they apply to a wide range of phenomena than when they apply only to a small piece of data.

3. *If you can't solve a problem, enlarge it*. A broad scope, as above, can be challenging, but it can also make things easier. Thus Dwight D. Eisenhower is reputed to have said: "If you can't solve a problem, enlarge it", meaning that putting a problem in a broader context may make it easier to solve. Good solutions to a problem may be hard to see

when the problem is viewed through a keyhole, but become visible when the door is opened.

4. *Micro-theories rarely generalise well.* Apart from the potential value of 'enlarging' a problem (point 2 above), and broad scope (point 1), a danger of adopting a narrow scope is that any micro-theory or theories that is developed for that narrow area are unlikely to generalise well to a wider context—with correspondingly poor results in terms of Simplicity and Power (see also Appendix B.6).

5. *Bottom-up strategies and the fragmentation of research.* The prevailing view about how to reach AGI seems to be "... that we'll get to general intelligence step by step by solving one problem at a time." expressed by Ray Kurzweil ([1], p. 234). And much research in AI has been, and to a large extent still is, working with this kind of bottom-up strategy: developing ideas in one area, and then trying to generalise them to another area, and so on.

   But it seems that in practice the research rarely gets beyond two areas, and, as a consequence, there is much fragmentation of research (see also Appendix B.6).

## 20. How to Overcome the Limited Scope for Adaptation in Deep Neural Networks

The problem considered here is that, contrary to how DNNs are normally viewed, they are relatively restricted in their scope for learning.

An apparent problem with DNNs is that, unless many DNNs are joined together [23], each one is designed to learn only one concept, which contrasts with the way that people learn multiple concepts, and these multiple concepts are often in hierarchies of classes or in part-whole hierarchies. Also, learning a concept by a DNN means only learning to recognise something like a cat or a house within a picture, with limited compositionality compared with the SPCM (Section 14).

In the SPCM, the concept of an SP-pattern, with the concept of SPMA, provides much greater scope for modelling the world than the relatively constrained framework of DNNs. This is because:

- Each concept in the SPTI is represented by one SP-pattern which, as a single array of SP-symbols, would normally be much simpler than the multiple layers of a DNN, with multiple links between layers, normally many than in knowledge structures created by the SPCM.
- There is no limit to the number of ways in which a given SP-pattern can be connected to other SP-patterns within SPMAs, in much the same way that there is no limit to the number of ways in which a given web page can be connected to other web pages.
- Together, these features of the SPTI provide much greater scope than with DNNs for the representation and learning of many concepts and their many inter-connections.

## 21. How to Eliminate the Problem of Catastrophic Forgetting

"We find that the CF [catastrophic forgetting] effect occurs universally, without exception, for deep LSTM-based [Long Short-Term Memory based] sequence classifiers, regardless of the construction and provenance of sequences. This leads us to conclude that LSTMs, just like DNNs [Deep Neural Networks], are fully affected by CF, and that further research work needs to be conducted in order to determine how to avoid this effect (which is not a goal of this study)." Monika Schak and Alexander Gepperth [46].

Catastrophic forgetting is the way in which, when a given DNN has learned one thing and then it learns something else, the new learning wipes out the earlier learning (see, for example, [47]). This problem is quite different from human learning, where new learning normally builds on earlier learning, as described in Section 8—although of course we all have a tendency to forget some things.

The SPCM is entirely free of the problem of catastrophic forgetting. The reasons that, in general, DNNs suffer from catastrophic forgetting and that the SPTI does not, are that:

- In DNNs there is a single structure for the learning and storage of new knowledge, a concept like 'my house' is encoded in the strengths of connections between artificial neurons in that single structure, so that the later learning of a concept like 'my car' is likely to disturb the strengths of connections for 'my house';
- By contrast, the SPCM has an SP-pattern for each concept in its repository of knowledge, there is no limit to the number of such SP-patterns that can be stored (apart from the limit imposed by the available storage space in the computer and associated information storage), and, although there can be many connections between SP-patterns, there is no interference between any one SP-pattern and any other.

However, it is possible with DNNs to sidestep the problem of catastrophic forgetting:

- As noted in Section 8, one may make a copy of a DNN that has already learned something, and then train it on some new concept that is related to what has already been learned. The prior knowledge may help in the learning of the new concept.
- If, for example, one wishes to train a DNN in, say, 50 new concepts, one can do it with a giant DNN that has space allocated to each of the 50 new concepts, each with multiple layers. Then providing that the training data for each concept is applied at the appropriate part of the giant DNN, there should be no catastrophic forgetting [23].

Arguably, these two solutions are ugly, without the elegance of the way the SPCM can, within the available storage space, learn multiple concepts smoothly, without any special provision, with transfer learning where required, and with the automatic building of networks of inter-related concepts where the data dictate it.

In addition, the proposed solutions for DNNs are likely to play havoc with calculations of the storage space in the human brain ([2], Section 11.4.9). This is because the SPCM requires only one SP-pattern for each concept, and provides for the sharing of structures for efficient compression. By contrast, a DNN requires multiple layers for each concept and the sharing of structures is likely to be absent or, at best, difficult.

## 22. Conclusions

This paper describes 20 significant problems in AI research, with potential solutions in the SP Theory of Intelligence.

In view of that potential, and with other evidence of the versatility of the SPTI in AI-related and non-AI-related applications (Appendix B), there are reasons to believe, as noted in the Introduction, that *the SPTI provides a relatively firm foundation for the development of human-level broad AI, aka artificial general intelligence*.

In addition to its being a promising FDAGI, the SPTI has potential as a foundation for a theory of human learning, perception, and cognition (Section 1.1).

There is also potential in the SPTI as a foundation for a theory of mathematics, logic, and computing (Section 1.1), a view which is radically different from the other 'isms' in the foundations of mathematics, and radically different from current concepts in logic and computing.

## 23. Software Availability

The source code and Windows executable code for the SP Computer Model is available via links under the heading "SOURCE CODE" on this web page: tinyurl.com/3myvk878.

The software is also available as 'SP71', under 'Gerry Wolff', in Code Ocean (codeocean.com/dashboard).

**Funding:** This research received no external funding.

**Data Availability Statement:** No new data were generated or analysed in support of this research.

**Acknowledgments:** I am very grateful to anonymous reviewers for constructive comments on earlier drafts of this paper.

**Conflicts of Interest:** The authors declare no conflict of interest.

**Abbreviations**

Abbreviations used in this paper are detailed here.

AI        Artificial Intelligence
AGI       Artificial General Intelligence
CSRK      Commonsense Reasoning and Commonsense Knowledge
DNN       Deep Neural Network
FDAGI     Foundation for the Development of AGI
HLPC      Human Learning, Perception, and Cognition
IC         Information Compression
ICMUP     Information Compression via the Matching and Unification of Patterns
NL        Natural Language
SPCM      SP Computer Model
SPMA      SP-multiple-alignment
SPTI       SP Theory of Intelligence

**Appendix A. High Level View of the SPTI**

The SPTI and its realisation in the SPCM is introduced here. This should be sufficient for understanding the rest of the paper.

But if more information is needed, the SPTI is described quite fully in [3], and even more fully in the book *Unifying Computing and Cognition* [2]. These and other publications in the SP programme of research are detailed on tinyurl.com/2p88zwr3 (accessed on 2 September 2022), most of them with download links.

In this research, the SPCM gives precision to concepts in the SPTI, it has been an invaluable means of testing ideas for the workings of the SPTI, and it is a means of demonstrating what can be done with the system. The SPTI itself is everything in the SPCM plus verbal descriptions and diagrams.

In broad terms, the SPTI is a brain-like system that takes in *New* information through the system's senses, compresses it, and stores it in a repository of *Old* information. This is shown schematically in Figure A1.

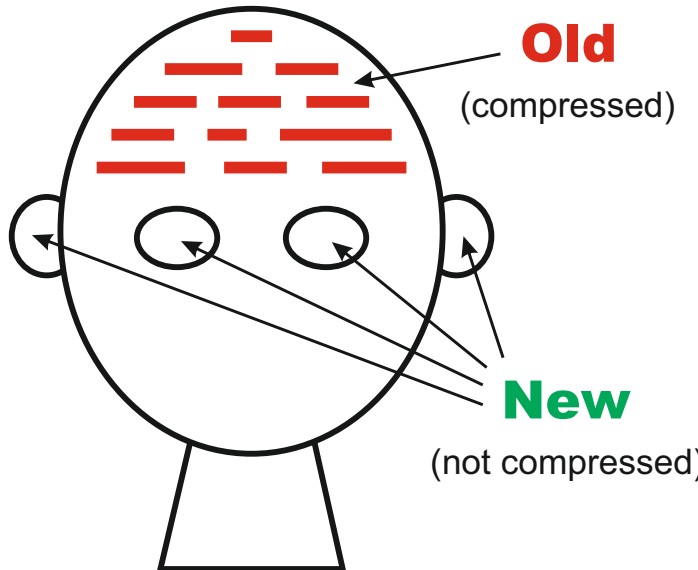

**Figure A1.** Schematic representation of the SPTI from an 'input' perspective. Reproduced from Figure 1 in [3].

For any one batch of New information, compression is achieved via the detection and elimination of redundancies *within* the given batch of New information, and also *between* the New information and pre-established Old information.

In the SPTI, all kinds of knowledge are represented with *SP-patterns*, where each such SP-pattern is an array of atomic *SP-symbols* in one or two dimensions. An SP-symbol is simply a 'mark' that can be matched with any other SP-symbol to determine whether the two SP-symbols are the same or different.

At present, the SPCM works only with one-dimensional SP-patterns. The addition of SP-patterns in two dimensions should open up new possibilities as sketched in Section 18.

### Appendix A.1. Information Compression and Intelligence

As indicated above, IC is central in all kinds of processing in the SPCM and in the structuring of knowledge within the SPCM.

The motivation for giving IC such an important role in the SPTI is an accumulation of evidence for the importance of IC in HLPC (Appendix B.4).

In broad-brush terms, all IC in the SPCM is achieved via a search for patterns that match each other and the merging or *unification* of patterns that are the same. Here, partial matches between patterns are as important as full matches.

More precisely, IC is achieved via the powerful concept of *SP-multiple-alignments* (SPMAs) (Appendix A.3), and via the unsupervised learning of SP-grammars (Section 5).

There is further discussion of ICMUP in Appendix B.4.2.

### Appendix A.2. Origin of the Name 'SP'

Since people often ask, the name 'SP' originated like this:

- The SPTI aims to simplify and integrate observations and concepts across a broad canvass (Section 19), which means applying IC to those observations and concepts;
- IC is a central feature of the structure and workings of the SPTI itself (Appendix A.1);
- And IC may be seen as a process that increases the *Simplicity* of a body of information, **I**, whilst retaining as much as possible of the descriptive and explanatory *Power* of **I**.

It is intended that 'SP' should be treated as a name, without any need to expand the letters in the name, as with such names as 'IBM' or 'BBC'.

### Appendix A.3. SP-Multiple-Alignment

The most important part of the SPCM is the software for creating SP-multiple-alignments (SPMAs), an example of which is shown in Figure A3.

The SPMA concept is largely responsible for the strengths of the SPTI in AI-related functions, summarised in Appendix B.1, for the potential benefits and applications of the SPTI in other areas, summarised in Appendix B.2, and for the clear potential of the SPTI to solve 20 significant problems in AI research, as described in this paper.

Bearing in mind that underplaying the advantages of a system is just as bad as overselling the system's advantages, it seems fair to say that *the concept of SP-multiple-alignment may prove to be as significant for an understanding of 'intelligence' as is DNA for biological sciences. SP-multiple-alignment may prove to be the 'double helix' of intelligence.*

### Appendix A.3.1. The SPMA Concept Is Inspired by and Derived from the Bioinformatics Concept of 'Multiple Sequence Alignment'

The concept of SPMA is here introduced via the concept from which it was derived: the bioinformatics concept of 'multiple sequence alignment', illustrated in Figure A2.

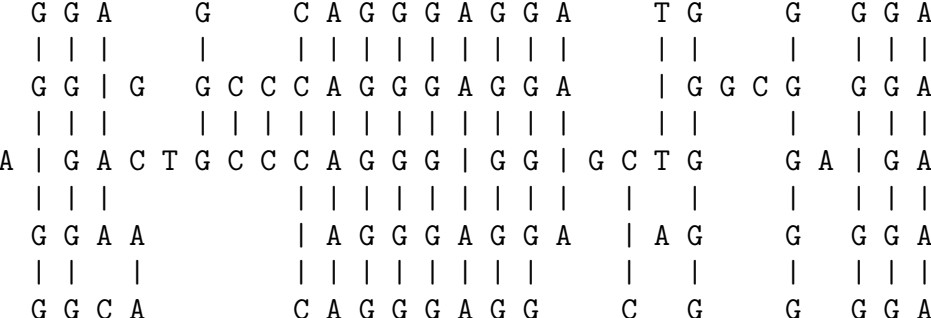

**Figure A2.** A 'good' multiple sequence alignment amongst five DNA sequences.

In the figure, there are five DNA sequences which have been arranged alongside each other, and then, by judicious 'stretching' of one or more of the sequences in a computer, symbols that match each other across two or more sequences have been brought into line.

A 'good' multiple sequence alignment, like the one shown (Figure A2), is one with a relatively large number of matching symbols from row to row.

The process of discovering a good multiple sequence alignment is normally too complex to be done by exhaustive search, so heuristic methods are needed, building multiple sequence alignments in stages and, at each stage, selecting the best partial structures for further processing.

Appendix A.3.2. How SPMAs Differ from Multiple Sequence Alignments

An SPMA (Figure A3) differs from a multiple sequence alignment like this:

- Each row contains one SP-pattern;
- The top row is normally a single New SP-pattern, newly received from the system's environment. Sometimes there is more than one New SP-pattern in row 0.
- Each of the other rows contain one Old SP-pattern, drawn from a store which normally contains many Old SP-patterns;
- A 'good' SPMA is one where the New SP-pattern(s) may be encoded economically in terms of the Old SP-patterns, as explained in ([3], Section 4.1).

As with multiple sequence alignments, heuristic methods must be used in the creation of SPMAs.

AI-related and non-AI-related strengths of the SPTI, due largely to the SPMA construct, are summarised in Appendix B.

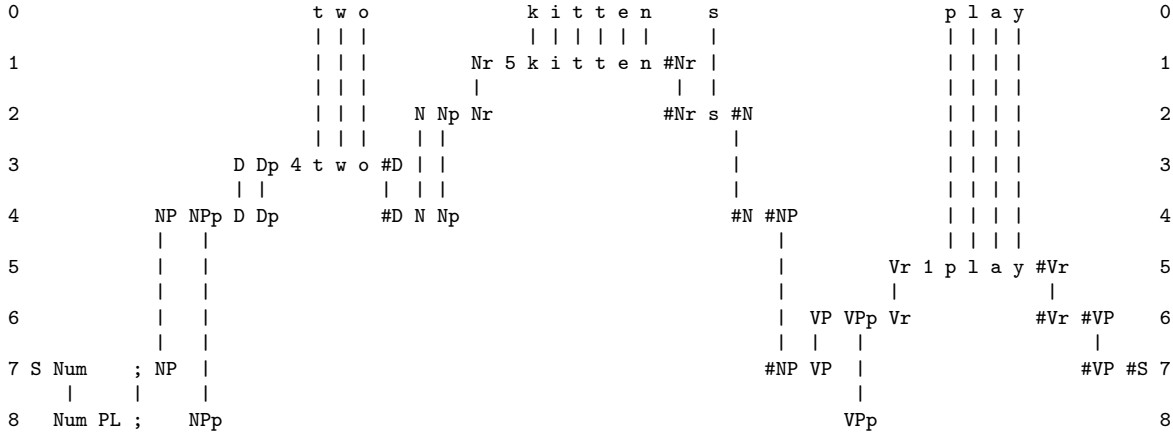

**Figure A3.** The best SPMA created by the SPCM with a store of Old SP-patterns like those in rows 1 to 8 (representing syntactic structures, including words) and a New SP-pattern, 't w o k i t t e n s p l a y', shown in row 0, representing a sentence to be parsed. The SP-pattern in row 8 is concerned with syntactic dependencies as described in Section 4.1.2. Adapted from Figure 1 in [20].

*Appendix A.4. Unsupervised Learning in the SPCM*

Apart from the SPMA construct, the second main part of the SPCM is procedures for the unsupervised learning of new knowledge. For any given body of New information, unsupervised learning means the creation of one or more 'good' *SP-grammars* via IC, where an SP-grammar is a set of Old SP-patterns that are relatively good for the economical encoding of that New information.

Unsupervised learning is a process that searches for redundancies within the New SP-pattern(s), and between the New SP-pattern and the repository of Old SP-patterns, and unifies redundancies that have been found, thus achieving IC.

As with the building of multiple sequence alignments and SP-multiple-alignments, the process of creating good SP-grammars cannot be achieved by exhaustive search. Heuristic methods are needed, building each SP-grammar in stages and discarding all but the best partial SP-grammar at the end of each stage.

There is more information in Section 5.

*Appendix A.5. Future Developments*

It is envisaged that the SPCM will be developed into an *SP Machine* with high levels of parallel processing and an improved user interface. This will facilitate further developments by researchers anywhere in the world, along the lines described in [15].

Figure A4, shows a schematic representation of how the SP Machine may be developed and applied.

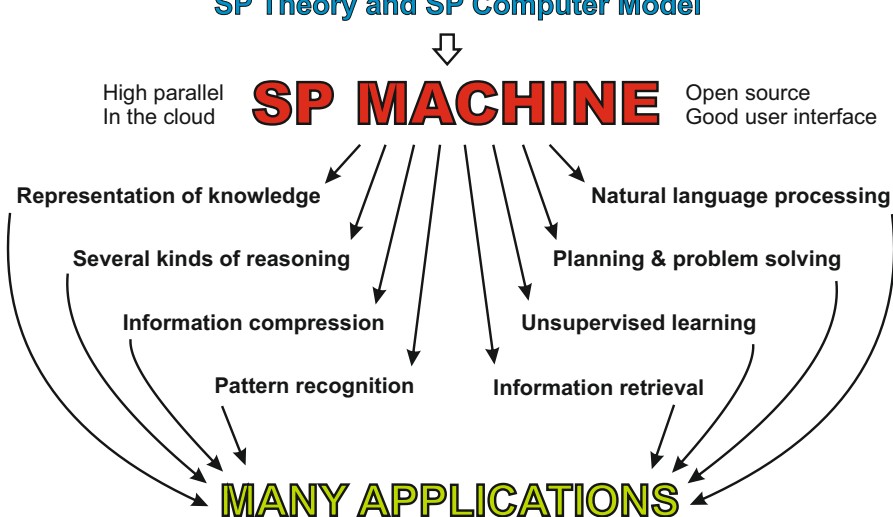

**Figure A4.** Schematic representation of the development and application of the SP Machine. Reproduced from Figure 2 in [3].

## Appendix B. Strengths of the SPTI in AI and Beyond

The strengths of the SPTI, in AI-related functions and beyond, are summarised in this appendix. Further information may be found in ([3], Sections 5 to 12), in ([2], Chapters 5 to 9), and in other sources referenced below.

Most of the AI-related strengths of the SPTI, described in Appendix B.1, are demonstrable with the SPCM. But the word 'strengths' is also applied to the potential of the SPTI to solve problems described in the body of this paper, and in Appendices B.2 and B.4, and to broad features of the SP research, as described in Appendices B.5 to B.7.

*Appendix B.1. AI-Related Strengths of the SPCM*

What are mainly demonstrable strengths of the SPCM are summarised in this section, with indications of the few cases where a capability is potential, not demonstrated. Many examples of the workings of the SPCM are given in [2,3].

Appendix B.1.1. Several Kinds of Intelligent Behaviour

The SPCM has demonstrable strengths in the following aspects of intelligence: unsupervised learning, including the discovery of segmental structures and classes of such structures; the analysis and production of NL; pattern recognition that is robust in the face of errors in data; pattern recognition at multiple levels of abstraction; computer vision [19]; best-match and semantic kinds of information retrieval; several kinds of reasoning (next subsection); planning; and problem solving.

Appendix B.1.2. Several Kinds of Probabilistic Reasoning

Because of the intimate relation between IC and concepts of inference and probability (Section 11), and owing to the fundamental role of IC in the workings of the SPTI, the system is inherently probabilistic. Accordingly, it is relatively straightforward for the SPCM to calculate absolute and relative probabilities for all aspects of intelligence exhibited by the SPCM. Details of those calculations are given in ([3], Section 4.4) and ([2], Section 3.7).

Kinds of reasoning that may be exhibited by the SPCM include: one-step 'deductive' reasoning; chains of reasoning; abductive reasoning; reasoning with probabilistic networks and trees; reasoning with 'rules'; nonmonotonic reasoning and reasoning with default values; Bayesian reasoning with 'explaining away'; causal reasoning; reasoning that is not supported by evidence; the inheritance of attributes in class hierarchies; and inheritance of contexts in part-whole hierarchies (([3], Section 10), ([2], Chapter 7)).

There is also potential in the system for spatial reasoning ([44], Section IV-F.1), and for what-if reasoning ([44], Section IV-F.2).

Appendix B.1.3. The Representation and Processing of Several Kinds of
AI-Related Knowledge

Although SP-patterns are not very expressive in themselves, they come to life in the SPMA framework within the SPCM. Within the SPMA framework, they provide relevant knowledge for each aspect of intelligence mentioned in Appendix B.1.1, for each kind of reasoning mentioned in Appendix B.1.2, and more.

More specifically, they may serve in the representation and processing of such things as: the syntax of NLs; class-inclusion hierarchies (with or without cross classification); part-whole hierarchies; discrimination networks and trees; if-then rules; entity-relationship structures ([20], Sections 3 and 4); relational tuples (([2], Chapter 10), Section 3), and concepts in mathematics, logic, and computing, such as 'function', 'variable', 'value', 'set', and 'type definition' (([2], Chapter 10), ([43], Section 6.6.1), ([48], Section 2)).

As previously noted (Appendix A), the addition of two-dimensional SP patterns to the SPCM is likely to expand the capabilities of the SPTI to the representation and processing of structures in two-dimensions and three-dimensions, and the representation of procedural knowledge with parallel processing.

Appendix B.1.4. The Seamless Integration of Diverse Aspects of Intelligence, and Diverse Kinds of Knowledge, in Any Combination

An important additional feature of the SPCM, alongside its versatility in aspects of intelligence and diverse forms of reasoning, and its versatility in the representation and processing of diverse kinds of knowledge, is that *there is clear potential for the SPCM to provide for the seamless integration of diverse aspects of intelligence and diverse forms of knowledge, in any combination.* This is because those several aspects of intelligence and several

kinds of knowledge all flow from a single coherent and relatively simple source: the SPMA framework.

It appears that this kind of seamless integration is *essential* in any artificial system that aspires to human-level broad intelligence.

Figure A5 shows schematically how the SPTI, with SPMA at centre stage, exhibits versatility and seamless integration.

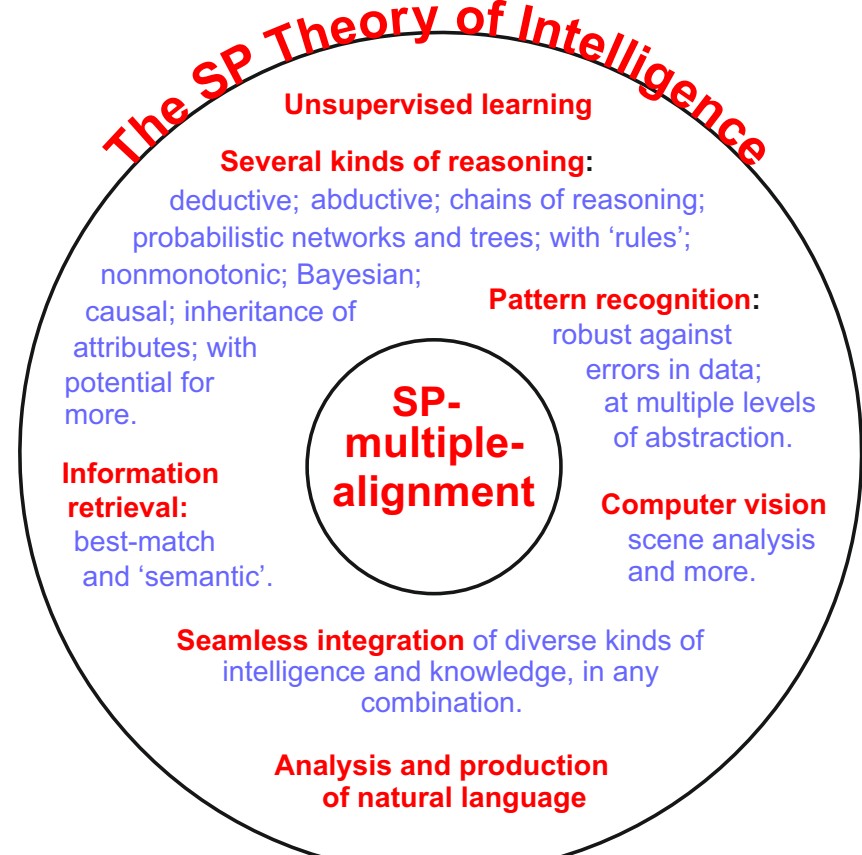

**Figure A5.** A schematic representation of versatility and seamless integration in the SPTI, with the SPMA concept centre stage.

Appendix B.1.5. How to Make Generalisations without over- or under-Generalisation; and How to Minimise the Corrupting Effect of 'Dirty Data'

The central role of IC in the workings of the SPCM (Appendixes A.1 and B.4) provides what appears to be a sound solution to two problems with unsupervised learning: how to generalise beyond a body of data (**I**) without either over-generalisations (under-fitting) or under-generalisations (over-fitting); and how to learn correct forms despite the fact that **I** normally contains errors of various kinds, otherwise called 'dirty data'.

The proposed solution, indebted to Ray Solomonoff [27,28], is described in ([3], Section 5.3) and ([2], Section 2.2.12). In brief: compress **I** as thoroughly as possible via unsupervised learning to yield an SP-grammar (**G**), and an encoding (**E**) of **I** in terms of **G**. Then discard **E** which contains all of the dirty data or most of it, and retain **G** which provides a compact description of **I**, including 'correct' generalisations from **I**.

Informal tests with unsupervised learning in the SPCM, and also in the MK10 and SNPR computer models of language learning [14], suggest that these principles are sound, including the exclusion of over- and under-generalisations, and the learning of 'correct' forms without corruption by 'dirty data'.

*Appendix B.2. Other Potential Benefits and Applications of the SPTI*

Apart from the foregoing distinctive features and advantages of the SPTI, it has several other potential benefits and applications. Relevant publications are outlined below:

- *Overview of potential benefits and applications.* Several potential areas of application of the SPTI are described in [43]. The ones that relate fairly directly to AI include: best-match and semantic forms of information retrieval; the representation of knowledge, reasoning, and the semantic web.
- *The development of intelligence in autonomous robots.* The SPTI opens up a radically new approach to the development of intelligence in autonomous robots [44].
- *Commonsense reasoning and commonsense knowledge.* Largely because of research by Ernest Davis and Gary Marcus (see, for example, [49]), the challenges in this area of AI research are now better known. Preliminary work shows that the SPTI has promise in this area [34].
- *An intelligent database system.* The SPTI has potential in the development of an intelligent database system with several advantages compared with traditional database systems [20].
- *Medical diagnosis.* The SPTI may serve as a vehicle for medical knowledge and to assist practitioners in medical diagnosis, with potential for the automatic or semi-automatic learning of new knowledge [50].
- *NL processing.* The SPTI has strengths in the processing of NL (([3], Section 8), ([2], Chapter 5)).
- *Sustainability.* The SPTI has clear potential for substantial reductions in the very large demands for energy of standard DNNs, and applications that need to manage huge quantities of data such as those produced by the Square Kilometre Array [51]. Where those demands are met by the burning of fossil fuels, there would be corresponding reductions in the emissions of $CO_2$.
- *Transparency in computing.* By contrast with applications with DNNs, the SPTI provides a very full and detailed audit trail of all its processing, and all its knowledge may be viewed. Also, there are reasons to believe that, when the system is more fully developed, its knowledge will normally be structured in forms that are familiar such as class-inclusion hierarchies, part-whole hierarchies, run-length coding, and more. Strengths of the SPTI in these area are described in [26].
- *Vision, both artificial and natural.* The SPTI opens up a new approach to the development of computer vision and its integration with other aspects of intelligence, and it throws light on several aspects of natural vision: [19,52].

*Appendix B.3. The Clear Potential of the SPTI to Solve 20 Significant Problems in AI Research*

As described in this paper, the potential of the SPTI to solve 20 significant problems in AI research is a substantial addition to evidence in support of the SPTI.

*Appendix B.4. Evidence for the Importance of IC in HLPC Suggests That IC Should Be Central in the SPCM*

A potent idea, pioneered by Fred Attneave [53,54], Horace Barlow [55,56], and others, is that much of the workings of brains and nervous systems may be understood as IC. This idea has been investigated by various researchers up to the present (see, for example, [57–59]). And the importance of IC in HLPC became central in a programme of research developing computer models of the learning of a first language by children [14]. Evidence for the importance of IC in HLPC is reviewed in [18].

In connection with this research and the quest for AGI, it is of interest that, as far back as 1969, Barlow wrote:

"... the operations needed to find a less redundant code have a rather fascinating similarity to the task of answering an intelligence test, finding an appropriate scientific concept, or other exercises in the use of inductive reasoning. Thus,

redundancy reduction may lead one towards understanding something about the organization of memory and intelligence, as well as pattern recognition and discrimination." ([56], p. 210).

where "find[ing] a less redundant code" leads to "redundancy reduction" which means IC. With regard to the goal of developing AGI:

- Evidence for the importance of IC in HLPC has provided the motivation for making IC central in the structure and workings of the SPCM;
- In view of the same evidence, it seems clear that IC should be central in the workings of any system that aspires to AGI;
- The central role for IC in the SPCM—mediated by the concept of SPMA (Appendix A.3)—is largely responsible for the strengths of the SPTI (Appendix B);
- In both natural and artificial systems:

  - For a given body of information, **I**, to be stored, IC means that a smaller store is needed. Or for a store of a given capacity, IC facilitates the storage of a larger **I** ([18], Section 4);
  - For a given body of information, **I**, to be transmitted along a given channel, IC means an increase in the speed of transmission. Or for the transmission of **I** at a given speed, IC means a reduction in the bandwidth which is needed ([18], Section 4).

- Because of the intimate relation between IC and concepts of inference and probability (Section 11), and because of the central role of IC in the SPTI, the SPTI is intrinsically probabilistic.

  Correspondingly, it is relatively straightforward to calculate absolute and relative probabilities for all aspects of intelligence exhibited by the SPTI, including several kinds of reasoning (([3], Section 4.4), ([2], Section 3.7)), in keeping with the probabilistic nature of human inferences and reasoning.

Appendix B.4.1. A Resolution of the Apparent Paradox That IC May Achieve Decompression as Well as Compression of Data

It is sometimes said that IC as a central feature of HLPC conflicts with the undoubted fact that people can and do produce information as well as compress it, both in ordinary speech or writing and also in creative areas like creative writing, painting, the composition of music, and so on.

In that connection, an interesting feature of the SPCM is that SPMA processes for the analysis of New information are *exactly* the same as may be used for the production of information. For example, with NL, processes for the production of a sentence are, without any qualification, the same as may be used for the analysis of the same sentence.

Since the SPCM works by compressing information, this feature of the SPCM looks, paradoxically, like "decompression of information by compression of information".

How the whole system works, and how this paradox may be resolved, is explained in ([3], Section 4.5) and ([2], Section 3.8).

There is clear potential in the SPCM for the creation of entirely new structures which may be seen as novel or creative, but not necessarily artistic. This is an aspect of the SPTI that is waiting to be explored.

Appendix B.4.2. The Working Hypothesis That IC May Always Be Achieved via the Matching and Unification of Patterns

A working hypothesis in the SP research is that all kinds of IC may be understood as ICMUP.

Although this is a "working hypothesis", there is much supporting evidence: the powerful concept of SPMA may be understood as an example of ICMUP [60]; the SPMA construct seems to underpin several aspects of intelligence (Appendix B.1), including sev-

eral kinds of probabilistic reasoning (Appendix B.1.2); and much of mathematics, perhaps all of it, may be understood in terms of ICMUP [6].

In this research, seven main variants of ICMUP are recognised ([6], Sections 5.1 to 5.7):

- *Basic ICMUP*. Two or more instances of any pattern may be merged or 'unified' to make one instance ([6], Section 5.1).
- *Chunking-with-codes*. Any pattern produced by the unification of two or more instances is termed a 'chunk'. A 'code' is a relatively short identifier for a unified chunk which may be used to represent the unified pattern in each of the locations of the original patterns ([6], Section 5.2).
- *Schema-plus-correction*. A 'schema' is a chunk that contains one or more 'corrections' to the schema. For example, a menu in a restaurant may be seen as a schema that may be 'corrected' by a choice of starter, a choice of main course, and a choice of pudding ([6], Section 5.3).
- *Run-length coding*. In run-length coding, a pattern that repeats two or more times in a sequence may be reduced to a single instance with some indication that it repeats, or perhaps with some indication of when it stops, or even more precisely, with the number of times that it repeats ([6], Section 5.4).
- *Class-inclusion hierarchies*. Each class in a hierarchy of classes represents a group of entities that have the same attributes. Each level in the hierarchy *inherits* all the attributes from all the classes, if any, that are above it ([6], Section 5.5).
- *Part-whole hierarchies*. A part-whole hierarchy is similar to a class-inclusion hierarchy but it is a hierarchy of part-whole groupings ([6], Section 5.6).
- *SP-multiple-alignment*. The SPMA concept ([6], Section 5.7) is described in Appendix A.3.

*The SPMA concept may be seen as a generalisation of the other six variants of ICMUP*, as demonstrated via the SPCM in [60].

This list probably does not exhaust the possible variants of ICMUP, but they are the ones that have received most attention so far in the SP programme of research.

*Appendix B.5. The SPTI Provides an Entirely Novel Perspective on the Foundations of Mathematics*

In view of evidence for the importance of IC in HLPC (Appendix B.4), and in view of the fact that mathematics is the product of human brains and has been designed to help human thinking, it should not be surprising to find that IC is central in the structure and workings of mathematics.

In keeping with that line of thinking, the concept of ICMUP provides an entirely novel perspective on the foundations of mathematics, described in the paper [6]. It is radically different from any of the existing 'isms' in the foundations of mathematics. There are potential connections with structuralism, except that structuralism has no place for IC or ICMUP ([6], Section 4.4.4) and it differs in many other ways from the SPTI.

*Appendix B.6. The Benefits of a Top-Down, Breadth-First Research Strategy with Wide Scope*

Although the SP research strategy is not, in itself, a feature of the SPTI, the wide scope of the SP research strategy, described in this appendix, is yielding breadth in the other strengths of the SPTI and these are largely intrinsic features of the SPTI.

Allen Newell was one of the first people to draw attention to the problems of fragmentation in cognitive science in his famous paper "You can't play 20 questions with nature and win" [61]. In that paper he exhorted researchers to tackle "a genuine slab of human behaviour" (p. 303), thus avoiding the weaknesses of micro-theories with limited scope for generalisation towards the description and explanation of phenomena in HLPC.

This thinking led to his book *Unified Theories of Cognition* [62] and a programme of research developing the Soar cognitive architecture [63], aiming for a unified theory of cognition.

This work chimes with Pamela McCorduck's description of fragmentation in AI:

"The goals once articulated with debonair intellectual verve by AI pioneers appeared unreachable ... Subfields broke off—vision, robotics, natural language processing, machine learning, decision theory—to pursue singular goals in solitary splendor, without reference to other kinds of intelligent behaviour." ([64], p. 417).

Later, she writes of "the rough shattering of AI into subfields ... and these with their own sub-subfields—that would hardly have anything to say to each other for years to come." ([64], p. 424).

She adds: "Worse, for a variety of reasons, not all of them scientific, each subfield soon began settling for smaller, more modest, and measurable advances, while the grand vision held by AI's founding fathers, a general machine intelligence, seemed to contract into a negligible, probably impossible dream." ([64], p. 424).

Although this quote is from 2004, much the same may be said today. There seems to be a widespread belief that, when a satisfactory system has been developed for one aspect of intelligence, it will be possible gradually to combine it with other systems in a bottom-up strategy leading to the full generality of AGI. And that belief—much the same as what Newell criticised in [61,62]—seems always to fail.

With these ideas and observations in mind, the SPTI, as noted in Section 19, has been developed via a top-down, breadth-first research strategy with an exceptionally wide scope, *aiming for a simplification and integration of observations and concepts across AI, mainstream computing, mathematics, and HLPC.*

The SP strategy should help to meet the concerns of Gary Marcus and Ernest Davis: "What's missing from AI today—and likely to stay missing, until and unless the field takes a fresh approach—is broad (or "general") intelligence." ([45], p. 15).

*Appendix B.7. The Benefits of a Biological Perspective in the Development of AGI*

In the same way that the SP research strategy (Appendix B.6) is yielding intrinsic strengths of the SPTI, much the same may be said about the biological perspective described in this appendix.

Since the main objective of the SP research is to develop a firm FDAGI, it is, arguably, clear that a biological perspective is likely to be helpful, where that perspective includes knowledge of cognitive psychology and neuroscience—since humans are biological entities, and since human intelligence is a biological phenomenon. More specifically, since human intelligence is the most fully developed intelligence on the planet, a knowledge of HLPC is likely to be beneficial in guiding research towards AGI. Without that knowledge, we are unnecessarily blindfolded in our research.

With that regard, the SP research has benefited in three main ways:

- Earlier research developing computer models of the unsupervised learning of a first language [14], mentioned elsewhere in this paper, has provided an inspiration and foundation for the development of the SPTI.
- Recognition of the importance of IC in HLPC (Appendix B.4), which depends on studies in psychology and neuroscience.
- The author of this paper, and the main driver in developing the SPTI, has a first degree from Cambridge University in the Natural Sciences Tripos, comprising studies in experimental psychology and other biology-related sciences.

## Appendix C. Definitions of Terms

Terms used in this paper are listed here with either or both of the section or appendix.

| | |
|---|---|
| SP-grammar | Section 5.1, Appendix A.4 |
| SP-multiple-alignment | Appendix A.3 |
| SP-pattern | Appendix A |
| SP-symbol | Appendix A |
| unification | Section 18, Appendix A.1 |

As noted in Appendix A.2, it is intended that 'SP' should be treated as a name, without any need to expand the letters in the name, as with such names as 'IBM' or 'BBC'.

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
