# Peer review of "Twenty Significant Problems in AI Research, with Potential Solutions via the SP Theory of Intelligence and Its Realisation in the SP Computer Model"

_foundations, doi:10.3390/foundations2040070_

Round 1

Reviewer 1 Report

Dear Professor,

it was a pleasure to review your work.

A few small clarifications:

Page 38 above -> "Appendix 18"? (wrong reference?)

Pag. 38 at the top of section A.3 -> "Appendix 19"? (wrong reference?)

Page 48 above -> "Appendix 11"? (wrong reference?)

I think your work is well done,

please add some industrial correlations:

HPLC in my opinion should be the basis of the perception and analysis of HCPSpace (Human Cyber ​​Physical Space).

https://doi.org/10.1016/j.eng.2019.11.014

same for...

SPTI for the NGIM (new generation of intelligent manufacturing).

https://doi.org/10.1016/j.eng.2018.01.002

Sincerely,

Author Response

Please see Responses to reviewer 1.docx, uploaded below.

Reviewer 2 Report

The proposed paper mainly summarizes previous work, mostly of the author himself. It is well structured and gives short descriptions of 20 problems in AI research and some possible solution approaches based on SPTI. 

Each of these approaches is briefly described, however formal, technical, and experimental details or a proof of the given claims, is missing. There are references to related work, where (hopefully) these formal foundation is provided (did not check as part of this review), but in this paper, such kind of information is missing.

In general, the paper looks more like a magazine article and not like a scientific publication. It does not provide any new scientific information.

Furthermore, the paper does not follow the layout guidelines given by the MDPI. 

Author Response

Pleaase see Responses to reviewer 2.docx uploaded below.

Round 2

Reviewer 2 Report

Dear Author,

thank you for your comments and the rework of your paper. One major problem - of course - is, that it is submitted in a completely different format than the typical MDPI-layout. But I hope, the colleagues from the editing team will correct this. 

Regarding your comments: 

Point 1 > due to the changes you made, particularly providing more evidence and insight, the paper gained more depth and more research-style. Thank you for this. 

Point 2 > I agree to your comment. 

Point 3 > the new version does no longer have this impression (self-promotion). Thank you for changing

Point 4 > Thank you, as well

Point 5 > I still think, the paper will be an excellent source for future work and will serve as a reference document for other researchers. However, it's unique contribution to the field mainly consists of collecting, summarizing, and outlining the topics (which is still very important). 

Point 6 > two stars were given due to the missing experiments and details. Revised now

Point 7 > changed to three stars

Point 8, 9 > I still have the same opinion. 

Point 10 > already discussed

Point 11 > changed to three star rating

Point 12 > thank you very much for the changes

Point 13 > already outlined.

Point 14 > see introductory comment